# Hyperspectral library of submerged aquatic vegetation and benthic substrates in the Baltic Sea

Ele Vahtmäe[1], Laura Argus[1], Kaire Toming[1], Martin Ligi[1], Tiit Kutser[1]

[1]Estonian Marine Institute, University of Tartu, Mäealuse 14, Tallinn, 12618, Estonia

*Correspondence to*: Ele Vahtmäe (ele.vahtmae@ut.ee)

**Abstract.** A hyperspectral reflectance database was acquired for the Baltic Sea submerged aquatic vegetation (SAV) and bare substrates by using Ramses (TriOS) radiometers capturing the spectral data within the visible (VIS) and near infrared (NIR) spectral range. The target samples included the most dominant and characteristic SAV species in the Baltic Sea, as well as several bare substrate types and beach cast communities. Target samples were measured within the 350 to 900 nm wavelength
range under sunlight conditions without the water column influence i.e. samples were taken out of the water. Such library is expected to provide insight into the spectral properties of various SAV species and substrates occurring in the coastal waters of the temperate geographic regions facilitating development of algorithms for differentiation and mapping various SAV communities. Additionally, measured reflectance spectra can be used as spectral endmembers in physical models and classification algorithms for coastal vegetation mapping and quantification. Data is openly available at PANGAE online
repository https://doi.pangaea.de/10.1594/PANGAEA.971518 (Vahtmäe et al., 2024).

## 1 Introduction

Vegetated coastal ecosystems provide valuable ecosystem services – constitute feeding, spawning, and sheltering grounds for wide variety of species, act as soft sediment stabilizers, protect coastline, remove nutrients and contaminants from the water column etc. (Cotas et al., 2023; Macreadie et al., 2017). They also play important role in climate change mitigation by
sequestering and storing carbon from the atmosphere (Duarte et al., 2005; McLeod et al., 2011). Information on such highly valued ecosystems is important from both scientific and management perspectives. Optical remote sensing can provide information on large temporal and spatial scales, which has led to increased use of such technology in coastal studies (Kutser et al., 2020). To better implement those technologies, there is a need for improved knowledge of spectral properties of vegetation species and benthic substrates inhabiting coastal areas.

The Baltic Sea is located in the temperate geographic region. It is semi-enclosed non-tidal water body, that lack intertidal zone and where benthic vegetation species mostly grow submerged. Submerged aquatic vegetation (SAV) in the Baltic Sea include several taxonomic groups of macroalgae, as well as higher plants e.g. vascular plants. Macroalgae are classified into three major groups: brown algae (*Phaeophyceae*), green algae (*Chlorophyta*), and red algae (*Rhodophyta*) according to their pigmentation (Vimala and Poonghuzhali, 2013). Consequently, there exist a need to generate substantiated dataset representing

reflectance spectra of SAV species from different taxonomic groups. Such dataset can be used to generate insights into the spectral properties of SAV species and substrates characteristic to the Baltic Sea, but also to the broader temperate geographic region. As such, the current database contributes to the global dataset of SAV reflectance spectra allowing further analysis, whether the spectral resolutions of current and future remote sensing sensors can discriminate broader SAV classes and/or species based on their spectral signatures.

SAV distribution maps are mostly created by using different image-based classification approaches, such as unsupervised and supervised classification (Bouvet et al., 2003; Fornes et al., 2006; Phinn et al., 2012; Roelfsema et al., 2013; Traganos and Reinartz, 2018). Such image-based methods require high amount of ground truth data or detailed expert knowledge of the area to train classification algorithms (Campbell et al., 2023). Alternative here, is to use signal-based classification approaches, where measured and/or modelled spectral libraries are used to interpret imagery (Kutser et al., 2006; Lesser and Mobley, 2007;

Vahtmäe and Kutser, 2013). The signal-based classification does not require simultaneous and continuous field surveys, instead given method requires availability of end-member spectral library. The data we propose in the current work is designed to be this kind of library for SAV classification applications.

Benthic reflectance spectra are also required parameters in physics-based forward models e.g., HydroLight model for natural waters, where they can be used together with inherent water optical properties in numerical simulations if measured data are

not sufficiently represented or lacking. Additionally, physics-based bio-optical inversion models (e.g., WASI-2D, BOMBER, IDA, HOPE) require benthic endmembers as input parameters to model benthic signatures through the water column. Then, modelled spectra are compared with measured spectra from remote sensing images and optimization algorithms can retrieve SAV distribution and abundance assessments from this comparison (Dekker et al., 2011; Fritz et al., 2019; Gege, 2014; Giardino et al., 2012; Hedley et al., 2009, 2018). The collected spectra can serve as endmembers in such bio-optical forward

and inversion models. Spectral signatures of benthic endmembers can also be used to assess the quality of water column correction on satellite/airborne images, as benthic spectra without the water column influence should resemble to the spectra in our spectral library.

To meet all the abovementioned needs, the current work aimed to collect a dataset of hyperspectral reflectance measurements from SAV species and substrate types that naturally occur in the coastal waters of the Baltic Sea. This spectral dataset was

collected by the research team of the Estonian Marine Institute, University of Tartu. Various coastal areas were visited in Estonia and Sweden in the Baltic Sea to collect reflectance spectra of the most characteristic and dominant SAV species and substrate types. A subset of this database has already been used for example by (Kotta et al., 2014), where statistical differences between reflectance spectra of SAV species were quantified and spectral regions, contributing the most to the statistical differences, defined.

Although we aimed to capture reflectance spectra of the most dominant and characteristic SAV species and substrate types present in the coastal waters of the Baltic Sea, the dataset presented here is not complete. More importantly, spectral properties of SAV species may vary depending on seasonality and environmental conditions e.g. decrease of the chlorophyll concentration during senescence/stress. As a result, there exists a considerable variation in pigment composition and quantity

among broad taxonomic groups and even within species (Kotta et al., 2014). In the future, we plan to complement the collected

dataset with additional species and/or substrate types using similar approach presented here. Moreover, we aim to collect reflectance spectra of various SAV species all throughout the vegetation period to capture seasonal changes in their reflectance spectra. Still, we believe that in the present form, the dataset may lead to several implications to current and future satellite missions.

Measured reflectance spectra of various SAV species are displayed in number of publications (Chao Rodríguez et al., 2017;

Davies et al., 2023; Dekker et al., 2005; Fyfe, 2003; Kutser et al., 2006, 2020; Olmedo-Masat et al., 2020). However, such information is often lacking in the data format, which would allow to re-use the measured data. By making current dataset available to other researchers, we hope to encourage them to do similar work and propose new algorithms for SAV detection and classification.

## 2 Materials and methods

## 2.1 Samples

This dataset has been collected over a period of 2 years (2011-2012) during field campaigns in the Baltic Sea coastal waters, in Estonia and Sweden. The set of target samples, presented in this database was divided into 6 groups: red macroalgae, green macroalgae, brown macroalgae, higher plants, bare substrates, and beach cast (Fig 1).

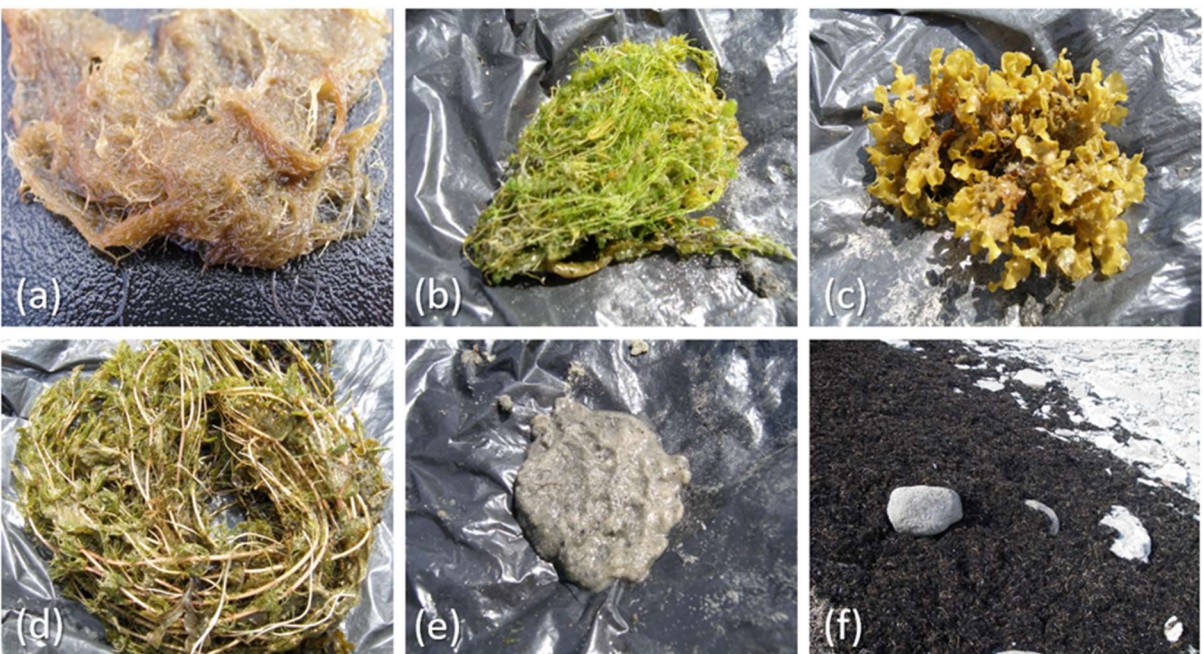

**Figure 1: Photographs of selected target samples of each of the 6 groups: (a) red macroalgae (*Ceramium tenuicorne*), (b) green macroalgae (*Chara spp.*), (c) brown macroalgae (*Fucus vesiculosus*), (d) higher plants (*Myriophyllum spicatum*), (e) bare substrate, (f) beach cast.**

Green macroalgae, red macroalgae and brown macroalgae are three major taxonomic groups of macroalgae in the Baltic Sea according to their pigmentation, each of which exhibits its own characteristic spectral features (groups 1-3, Table 1). In addition to macroalgae, the Baltic Sea also hosts higher plants or vascular plants (group 4, Table 1). Beside to the vegetated habitats, the benthic environment of the Baltic Sea includes unvegetated bare substrates (group 5, Table 1). Finally, the last group is beach cast, which consists of decaying vegetation material (group 6, Table 1). Several benthic vegetation species and substrate types were measured under each of the given six groups.

**Table 2: Measured SAV species and substrate types.**

|  | Species/types | Number of specimens |
|---|---|---|
| **Green macroalgae** (*Chlorophyta*) | *Cladophora glomerata* | 4 |
|  | *Chara spp.* | 6 |
|  | *Monostroma balticum* | 1 |
|  | *Ulva intestinalis* | 2 |
| **Red macroalgae** (*Rhodophyta*) | *Furcellaria lumbricalis* | 1 |
|  | *Ceramium tenuicorne* | 4 |
|  | *Polysiphonia fucoides* | 1 |
| **Brown macroalgae** (*Phaeophyceae*) | *Pilayella littoralis* | 3 |
|  | *Fucus vesiculosus* | 5 |
|  | *Dictyosiphon foeniculaceus* | 1 |
|  | *Chorda filum* | 1 |
| **Higher plants** | *Zannihellia palustris* | 1 |
|  | *Stuckenia pectinate* | 3 |
|  | *Myriophyllum spicatum* | 3 |
| **Bare substrate** | Sand | 4 |
|  | Pebble | 1 |
|  | Gravel | 1 |
|  | Limestone plate | 1 |
| **Beach cast** | Fresh beach-cast | 1 |
|  | Dry beach-cast | 2 |

Recorded red macroalgae species included *Furcellaria lumbricalis*, *Ceramium tenuicorne*, *Polysiphonia fucoides*; green macroalgae species included *Cladophora glomerata*, *Chara spp.*, *Monostroma balticum*, *Ulva intestinalis*; brown macroalgae

species included *Pilayella littoralis*, *Fucus vesiculosus*, *Dictyosiphon foeniculaceus*, *Chorda filum* and higher plant species included *Zannihellia palustris*, *Stuckenia pectinate*, *Myriophyllum spicatum* (Table 1). Bare substrate reflectance spectra are measured for sand, pebble, gravel, and limestone plate. Beach cast communities included SAV communities that were either recently washed out of the sea or already dried in the sun. Most of the SAV species and substrates are measured more than once (up to six different specimens) in different locations and/or during different field campaigns.

## 2.2 Spectral reflectance measurements

Before conducting measurements, all samples were taken out of the water and SAV species were identified to the lowest possible taxonomic level by the biologist. Most of the samples were identified to the species level, only charophytes from green macroalgae group were identified to their genus level (*Chara spp.*). All reflectance measurements were performed outdoor in the field, using solar light as illumination source. Samples were measured in the boat or on the beach either on the
natural background or placed on the artificial black background to minimize signal from the adjacent environment. Bare substrate samples (e.g., sand, gravel on the beach) and beach-casts were mostly measured at their location on the beach.

Ramses (TriOS GmbH, Germany) portable field radiometers were used to capture the spectral data within the 350 to 900 nm wavelength range with a spectral resolution of 3.3 nm. It is important to point out, that the signal to noise ratio (SNR) for the measurements below 400 nm and above 850 nm is significantly higher than within the rest of the spectral range. Ramses
measurement set consisted of two simultaneously operated sensors: irradiance and radiance sensors. The radiance sensor measured upwelling spectral radiance $L_u$ (W m$^{-2}$ nm$^{-1}$ sr$^{-1}$) and irradiance sensor measured downwelling spectral irradiance $E_d$ (W m$^{-2}$ nm$^{-1}$). Remote sensing reflectance ($R_{rs}$, sr$^{-1}$) was calculated as the ratio of $L_u/E_d$. As sensors measure at slightly different wavelengths, then before reflectance calculation, signals from both sensors were interpolated to a fixed wavelengths with a 2 nm step.

While the $E_d$ sensor was always attached to the standard TriOS measuring frame, the $L_u$ sensor was either attached to the frame or held in hand pointed down to the sample during measurements (Fig. 2). The field of view (FOV) of the Lu sensor is 7°, resulting in an imaged area of around 1.1 cm2 when positioned at a distance of 10 cm. For each sample, multiple consecutive measurements (5 to 10 individual measurements) were performed. All consecutively measured spectra were visually inspected, and outliers removed. After initial assessment, average spectra of multiple measurements were calculated for each sample to
reduce the noise.

For the current database Rrs was obtained for benthic species and substrates by using radiance and irradiance sensors. The spectral data can also be measured as reflectance (R), which is the ratio of upwelling radiance to downwelling radiance (Lu/Ld) or upwelling irradiance to downwelling irradiance (Eu/Ed). Often R is measured with only one sensor by using white spectralon panel as a reference (Chao Rodríguez et al., 2017; Davies et al., 2023; Fyfe, 2003; Olmedo-Masat et al., 2020). The relationship
between radiance and irradiance is not so straightforward, but in case of Lambertian surface, the radiance value can be multiplied by $\pi$ to get irradiance. Similarly, the outcome of the atmospheric correction, applied to the remote sensing imageries, can either be Rrs or R. If the outcome of the atmospheric correction is irradiance reflectance, then our Ramses measured Rrs

can be multiplied by the Q-factor, which converts it to the irradiance reflectance, making Ramses measurements thereafter comparable to the outcome of the atmospheric correction. The Q-value may range from 0.3 to 6.5 (Gentili and Morel, 1993), but to simplify it, the Q-factor can be considered equal to $\pi$.

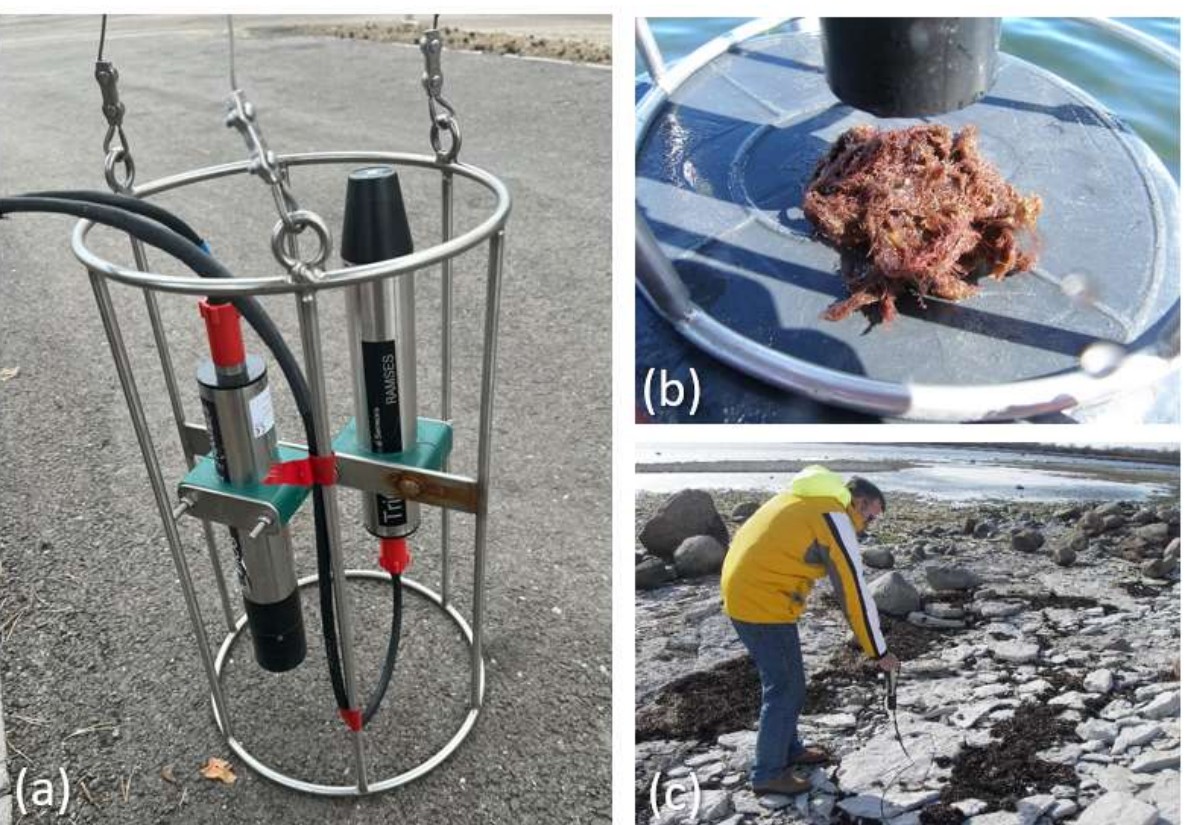

**Figure 2: Ramses two sensor measurement set (a), $L_u$ sensor installed to the measuring frame while conducting measurements (b), $L_u$ sensor held in hand while conducting measurements (c).**

## 3 Results

Spectral properties of SAV species are determined by their tissue morphology, cellular structure and the concentration and distribution of leaf biochemical components, such as pigments, water, nitrogen, cellulose and lignin pigment contents, shaping the formation of spectral signature of each specimen (Penuelas and Filella, 1998). Pigments absorb light at certain distinctive wavelengths, affecting the shapes of the reflectance spectra, while tissular structures mostly affect the absolute reflectance (Chao Rodríguez et al., 2017). All SAV groups contain chlorophyll-a (Chl-a), which is predominant green pigment in plants with absorption maximums in the blue (435 nm) and red (675 nm) part of the spectrum (Chao Rodríguez et al., 2017; Haxo

and Blinks, 1950). Therefore, all SAV groups in the current database exhibit characteristic vegetation spectra *e.g.*, low reflectance around 400-500 nm and 650-680 nm and a high reflectance in the near-infrared (NIR) spectral range (Fig. 3a-d).

Green macroalgae contain mostly chlorophyll pigments (Chl-a, Chl-b) (Chao Rodríguez et al., 2017; Rowan, 1989). As a result, green macroalgae show absorption minimums near 440 and 675 nm, which correspond to the chlorophylls absorption peaks, and a broad reflectance peak in green spectral range centred around 550 nm giving them characteristic green colour (Fig 3a). Chl-a is also present in brown and red macroalgae, but their green colour is partially masked out by other accessory pigments. Brown algae predominantly contain the brown pigment fucoxanthin, which absorbs light up to 560 nm (Rowan, 1989) shifting the reflectance maximums away from the green wavelengths further to the yellow spectral range. Typical spectral features of brown macroalgae are peaks around 600 and 650 nm and a shoulder around 575 nm (Fig. 3b). The red algae contain large quantities of the red pigment phycoerythrin, which absorbs light in the middle of the visible spectrum between 495 and 565 nm depressing also reflectance in the green part of the spectrum (Haxo and Blinks, 1950). Red macroalgae show two reflectance peaks in the red region of the visible spectrum, around 600 and 650 nm (Fig. 3c). Higher plants' characteristic pigments are similar to those of green macroalgae, showing higher reflectivity in green spectral range (Chao Rodríguez et al., 2017; Kutser et al., 2006, 2020). However, higher plant spectra (Fig. 3d) show flatter spectral shapes between 550 and 640 nm if compared to the green macroalgae spectra (Fig. 3a).

In contrast, bare substrates do not show specific red-edge reflectance or other pigment-induced spectral features in the visible wavelengths (Fig. 3e). The bare substrate group has the highest variability in absolute values, as this group includes substrate types with various brightness levels from muddy sand to bright limestone plate. The beach cast material in Fig. 3f show spectral characteristics different from both living SAV species and bare substrates. They exhibit very low signal in the visible spectral range and increasing reflectance in NIR spectral range. Beach-cast reflectance spectra were measured for fresh and dry beach-cast. While fresh beach-cast still had visible Chl-a absorption feature near 675 nm, then dry beach-cast had lost the chlorophyll absorption feature.

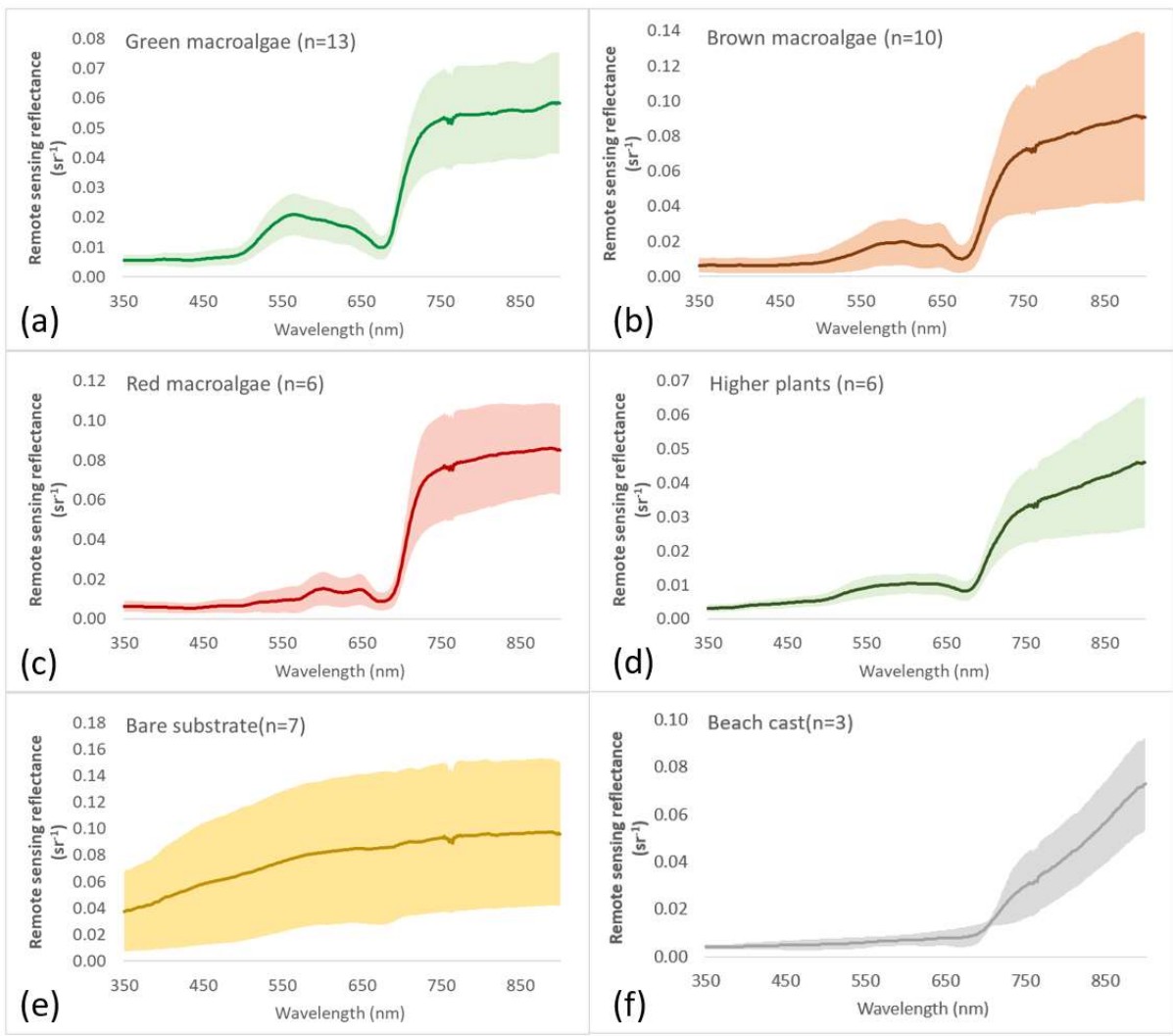

**Figure 3: Remote sensing reflectance (L_u/E_d) of six target groups: (a) green macroalgae, (b) brown macroalgae, (c) red macroalgae, (d) higher plants, (e) bare substrates, (f) beach cast. Mean spectra are represented with the solid lines and colour shading indicates the variability in standard deviation.**

## 4 Description of the dataset

This dataset contains the reflectance spectra of each target in 276 spectral bands (between 350 and 900 nm). Picture of the target was included if available. A metadata section is added to each spectral measurement containing the following information:

- Acquisition date.
- Country, where samples were collected.
- Location name.

- Latitude of measurement (approximate).
- Longitude of measurement (approximate).
- Illumination source.
- Measurement instrument name.
- Measurement unit.
- Use of reference panel and its name.
- Sampling environment (boat, beach, laboratory).
- Sampling background (natural, artificial).
- Target description (macrolagae, higher plants, substrate).
- Target species/genus name in *Latin.*

For every target sample, an average spectrum was calculated from 5-10 reflectance measurements. Averaging multiple measurements minimizes noise in the data.

## 5 Data availability

The current dataset is publicly available at PANGAE repository *https://doi.pangaea.de/10.1594/PANGAEA.971518* (Vahtmäe
et al., 2024).

## 6 Recommendations and conclusions

Remote sensing technology is increasingly used to detect, map, and monitor benthic ecosystems in shallow coastal waters. For the efficient implementation, the technology requires information about the spectral properties of benthic habitats. The database, presented here, aims to add new data on the spectral properties of the Baltic Sea benthic vegetation species enabling
to identify and characterize them and allowing to evaluate the potential to discriminate between them based on their spectral signatures. At the same time, it allows to facilitate comparative analysis of SAV species from different locations and regions all over the globe to study spectral variations within broader and narrower SAV groups. Additionally, reflectance spectra of several bare substrate types were recorded to facilitate discrimination between vegetated and non-vegetated areas.

Our hyperspectral reflectance database further improves scientific knowledge about optical characteristics of SAV species and
substrates. We believe such information is essential in remote sensing algorithm development and defining requirements for future remote sensing missions (spectral resolution, band selection, bandwidth, signal-to-noise ratio, etc.). The presented database can also be used in remote sensing applications, which require spectral information e.g., algorithm development, numerical simulations. The database can be used in remote sensing image processing, which require benthic endmembers (physics based radiative transfer modelling) and image classifications, where spectral libraries are used for classification.

## Author contribution

Conceptualization, EV; data curation, ML, LA; formal analysis, EV, KT; funding acquisition, TK; investigation, EV, LA, ML; visualisation, EV, KT; writing—original draft preparation, EV; writing—review and editing, KT, TK. All authors have read and agreed to the published version of the manuscript.

## Competing interests

The authors declare that they have no conflict of interest.

## Acknowledgement

We wish to thank the Department of Marine Biology in the Estonian Marine Institute for their help in performing field works and identifying the vegetation species.

## Financial support

This research was funded by the Estonian Research Council, grant number PRG2630.

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
