# Peer review of "Hyperspectral library of submerged aquatic vegetation and benthic substrates in the Baltic Sea"

_Earth System Science Data, 2024_

## Author Response (AR1)

**Reviewer 1**

Reviewer comment: This data paper describes a hyperspectral spectral library of aquatic vegetation from the Baltic Sea, with all samples taken out of water. The methodology for acquiring the spectra is well described, and all data can be seen and downloaded from the PANGAEA website. The rationale for such a dataset is justified: development of algorithms and requirements for future missions.

**Authors response**: We would like to thank the referee for the comments and for the positive feedback for the submitted data article.

**Reviewer comment: The authors could have cited the recent paper by Davies et al. 2023 RSE, who exploited a similar spectral library to test the loss of spectral resolution to discriminate the main macrophytes classes.**

**Authors response**: We agree that the paper by Davies et al. 2023 is worth of citing, as it presents spectral library of various benthic vegetation species/classes measured in the field.

**Changes in manuscript**: Page 3, line 77. The paper by Davies et al. 2023 was cited together with other papers, which present spectral libraries of benthic vegetation species: "Measured reflectance spectra of various SAV species are displayed in number of publications (Chao Rodríguez et al., 2017; Davies et al., 2023; Dekker et al., 2005; Fyfe, 2003; Kutser et al., 2006, 2020; Olmedo-Masat et al., 2020)..."

Reviewer comment: A graph and a small discussion of the spectral shapes of the main taxonomic groups related to the pigmentary composition provide helpful information for the reader. The number of specimens is sometimes limited to 1, which could have been improved.

**Authors response**: We also agree that the number of measured specimens remains sometimes low. We acknowledge that the published database is not complete and needs to be complemented with additional species and/or substrate types using similar approach presented in the paper. Still, we believe that in the present form, the dataset may lead to several implications to current and future satellite missions.

**Changes in manuscript**: Page 2, line 63. It was already acknowledged in the manuscript that the current dataset was far from being complete and we expressed hope that even in the present form, the dataset might lead to several implications to current and future satellite missions. Therefore, we did not make additional changes in the manuscript.

**Reviewer comment: It is not always clear if the species are subtidal or intertidal? Which fraction of the macrophytes biodiversity of the Baltic Sea is presented here?**

**Authors response:** In the current paper we targeted the most dominant and characteristic submerged aquatic species (SAV) in the Baltic Sea. The Baltic Sea is an enclosed non-tidal water body, therefore missing the intertidal zone. All the SAV species in the current paper mostly grow submerged (except some narrow coastal areas during low water level).

**Changes in manuscript**: Page 1, line 25. The manuscript was supplemented accordingly: "The Baltic Sea is located in the temperate geographic region. It is a semi-enclosed nontidal water body, that lack intertidal zone and where benthic vegetation species mostly grow submerged."

Reviewer comment: Some species' names should be checked, like Polysimphonia fucoides, which is probably Polysiphonia fucoides (but check WORMS as the latter name seems unaccepted).

**Authors response**: The referee is correct. The species name *Polysimphonia fucoides* needs to be corrected to *Polysiphonia fucoides*.

Changes in manuscript: The species name was corrected to Polysiphonia fucoides.

Reviewer comment: The Remote Sensing reflectance  $L_u/E_d$  is used here, but many spectral library papers use Reflectance Lu/Ld. The authors should comment on this and explicitly indicate how to shift from one to another.

Authors response: We had two simultaneously measuring Ramses (TriOS GmbH) sensors to capture the spectral data: irradiance sensor for measuring downwelling spectral irradiance (Ed) and radiance sensor for measuring upwelling spectral radiance (Lu). As a result, remote sensing reflectance (Rrs, sr-1) was calculated as the ratio of Lu/Ed. Referee is correct that spectral data can also be measured as radiance reflectance (R), which is the ratio of upwelling radiance to downelling radiance (Lu/Ld). Converting Ed to Ld is not so straightforward. In case of Lambertian surface Ed could be divided by a value of  $\pi$  to get Ld.

Changes in manuscript: Page 5, line 134. The following discussion about relationships between irradiance and radiance, as well as between remote sensing reflectance and irradiance reflectance was added:" For the current database Rrs was obtained for benthic species and substrates by using radiance and irradiance sensors. The spectral data can also be measured as reflectance (R), which is the ratio of upwelling radiance to downelling radiance (Lu/Ld) or upwelling irradiance to downwelling irradiance (Eu/Ed). Often R is measured with only one sensor by using white spectralon panel as a reference (Chao Rodriguez et al. 2017, Davies et al. 2023, Fyfe et al. 2003, Olmedo-Masat et al. 2020). The relationship between radiance and irradiance is not so straightforward, but in case of Lambertian surface, the radiance value can be multiplied by  $\pi$  to get irradiance. Similarly, the outcome of the atmospheric correction, applied to the remote sensing imageries, can either be Rrs or R. If the outcome of the atmospheric correction is irradiance reflectance, then our Ramses measured Rrs can be multiplied by the Q-factor, which converts it to the irradiance reflectance, making Ramses measurements thereafter comparable to the outcome of the atmospheric correction. The Q-value may range from 0.3 to 6.5 (Gentili and Morel, 1993), but to simplify it, the Q-factor can be considered equal to π."

Reviewer comment: To conclude, it is a valuable data paper, and I recommend accepting it with minor revisions.

**Reviewer 2**

This work documented a hyperspectral library of submerged aquatic vegetation (SAV) of Baltic Sea area by collecting the reflectance of targets using a handheld radiometer under a natural light environment. The significance of this data was well justified, and procedure for collecting the data was detailed. Dataset has also already been open to the public.

**Authors response**: We would like to thank the referee for the comments, and we will try to response to his/her concerns.

Reviewer comment: My concerns of this work are: 1) six targets (Figure 1, Table 2) were measured, leading to six spectral signatures of these benthic habitats. It does not make sense that such a small dataset is called a hyperspectral library. It is a very small dataset. This dataset cannot represent the diverse spectral signatures within a species, and spectral confusion between habitats.

**Authors response**: The measured target samples were divided into the six broad groups: green macroalgae, red macroalgae, brown macroalgae, higher plants, bare substrate and beach cast. Green macroalgae, red macroalgae and brown macroalgae are three major taxonomic groups of macroalgae in the Baltic Sea according to their pigmentation, each of which exhibits its own characteristic spectral features (groups 1-3, Table 1). In addition to macroalgae, the Baltic Sea also hosts higher plants or vascular plants (group 4, Table 1). Beside to the vegetated habitats, the benthic environment of the Baltic Sea includes unvegetated bare substrates (group 5, Table 1). Finally, the last group is beach cast, which consists of decaying vegetation material (group 6, Table 1). Several benthic vegetation species and substrate types were measured under each of the given six groups. In the current paper we targeted the most dominant and characteristic submerged aquatic species (SAV) and substrate types in the Baltic Sea.

Such a library of benthic endmembers gives a basic understanding about the spectral features of most common species/bare substrate types occurring in the temperate geographic region and therefore should allow analysing spectral differences/similarities between various taxonomic groups and between species. Referee is correct that since the current dataset does not include high number of measured spectra on species level (mostly between 1-6 measurements for each species), then the current dataset does not really allow analysing within species spectral variance. However, detecting within-species variation often tends to remain out of the scope of the satellite based benthic mapping due to restrictions in satellite's spectral and spatial resolution.

We agree that the number of measured specimens remains sometimes low. We acknowledge that the published database is not complete and needs to be complemented with additional species and/or substrate types using similar approach presented in the paper. Still, we believe that publishing such a library of benthic endmembers in the present form, the dataset may lead to several implications to current and future satellite missions.

**Changes in manuscript**: Page 4, line 94. It was more clearly emphasised that we divided our targets into six groups (not six targets). Characterization of each of the six groups is given: "Green macroalgae, red macroalgae and brown macroalgae are three major taxonomic groups of macroalgae in the Baltic Sea according to their pigmentation, each of which exhibits its own characteristic spectral features (groups 1-3, Table 1). In addition

to macroalgae, the Baltic Sea also hosts higher plants or vascular plants (group 4, Table 1). Beside to the vegetated habitats, the benthic environment of the Baltic Sea includes unvegetated bare substrates (group 5, Table 1). Finally, the last group is beach cast, which consists of decaying vegetation material (group 6, Table 1). Several benthic vegetation species and substrate types were measured under each of the given six groups. In the current paper we targeted the most dominant and characteristic submerged aquatic species (SAV) and substrate types in the Baltic Sea. "

Reviewer comment: 2) it is difficult to apply this dataset for any remote sensing classification models for mapping SAV or benthic habitats because it is measured without considering water column while SAV is located under water. Water conditions, density of each SAV species, seasonality, and the healthy condition of SAV largely impact the spectral signatures observed by airborne or spaceborne sensors. It is almost impossible to link the measured spectral signatures with any optical imaging sensor products for mapping purpose because they are measured by assuming they are not SAV (above water).

**Authors response**: We cannot agree with the referee on the current matter. Each additional centimetre of water column has a great influence on the benthic endmember spectra. Other users would not be able to benefit on benthic spectra measured together with the water column influence, as those spectra would then only be characteristic to the given water depth and given water quality. We find that it is highly important to measure the spectral signatures of benthic endmembers without the influence of the water column. The collected spectra can then serve as endmembers in various biooptical forward (e.g. Hydrolight) and inversion (e.g. WASI-2D, BOMBER) models, where they can be used together with suitable inherent water optical properties and water depths in numerical simulations. Those modelled spectra can then be used for image classification.

**Reviewer response**: For concerns 2, after water column correction, the spectral signature of optical airborne or spaceborne sensors might be comparable to the library here.

**Authors response:** Yes, reviewer is correct. Given spectral signatures of benthic endmembers can also be used to assess the quality of water column correction on satellite/airborne images, as given benthic spectra without the water column influence should resemble to the spectra in our spectral library.

**Changes in manuscript**: Page 2, line 53. Based to the reviewer's remark, one more value of the given spectral library was highlighted: "Spectral signatures of benthic endmembers can also be used to assess the quality of water column correction on satellite/airborne images, as benthic spectra without the water column influence should resemble to the spectra in our spectral library."

Reviewer comment: 3) the natural light is often not strong enough to collect spectral reflectance of samples, leading to weak reflectance as demonstrated in Figure 3. If these targets are exposed to artificial light which is often used by ASD Spectroradiometer in lab environments, the spectral signatures will be very different from the reported here. In general, the dataset is very small, and its potential application is limited.

**Authors response**: Again, we cannot agree with the referee on the current matter. Optical remote sensing from satellites, aircrafts, drones etc. relays on the natural light source –

sun. All remote sensing applications in the natural environment are performed by using the light from the sun. That is why our spectral measurements were also performed under the natural sun light.

In the current paper, we measured the reflectance of the target samples – the radiance from the sample was divided by the irradiance from the light source. As a result, we acquired the reflectance from the target sample which is not dependent on the light source. Weak spectral signal is related to the darker colour of the target sample. The darker the target sample - the lower the reflectance. In contrary, the brighter the target sample, the higher the reflectance.

**Reviewer response**: For concerns 3, noted that airborne or spaceborne sensors using natural sun light but they have a larger IFOV, leading to stronger reflectance. However, the handheld sensor used in this work has a very small IFOV, leading to weak signal, as reflected in the spectral profile shown in the manuscript. Maybe can pull out a satellite sensor reflectance of SAV for the same project area and compare them.

**Authors response**: We had two simultaneously measuring Ramses (TriOS GmbH) sensors to capture the spectral data: irradiance sensor for measuring downwelling spectral irradiance (Ed, W m-2 nm-1) and radiance sensor for measuring upwelling spectral radiance (Lu, W m-2 nm-1 sr-1). As a result, remote sensing reflectance (Rrs, sr-1) was calculated as the ratio of Lu/Ed.

If the outcome of your atmospheric correction is irradiance reflectance (R), then measured remote sensing reflectance can be multiplied by the Q-factor, which converts it to the irradiance reflectance, making Ramses measurements thereafter comparable to the outcome of the atmospheric correction. The Q-value may range from 0.3 to 6.5 steradians, but to simplify it, the Q-factor can be considered equal to  $\pi$ .

Changes in manuscript: Page 5, line 134. The following discussion about relationships between irradiance and radiance, as well as between remote sensing reflectance and irradiance reflectance was added:" For the current database Rrs was obtained for benthic species and substrates by using radiance and irradiance sensors. The spectral data can also be measured as reflectance (R), which is the ratio of upwelling radiance to downelling radiance (Lu/Ld) or upwelling irradiance to downwelling irradiance (Eu/Ed). Quite often R is measured with only one sensor by using white spectralon panel as a reference (Chao Rodríguez et al., 2017; Davies et al., 2023; Fyfe, 2003; Olmedo-Masat et al., 2020). The relationship between radiance and irradiance is not so straightforward, but in case of Lambertian surface, the radiance value can be multiplied by  $\pi$  to get irradiance. Similarly, the outcome of the atmospheric correction, applied to the remote sensing imageries, can either be Rrs or R. If the outcome of the atmospheric correction is irradiance reflectance, then our Ramses measured Rrs can be multiplied by the Q-factor, which converts it to the irradiance reflectance, making Ramses measurements thereafter comparable to the outcome of the atmospheric correction. The Q-value may range from 0.3 to 6.5 (Gentili and Morel, 1993), but to simplify it, the Q-factor can be considered equal to  $\pi$ ."